# OpenReview forum: "On the Role of Reasoning Traces in Large Reasoning Models"
_ICLR.cc/2026/Conference — ICLR 2026 Conference Withdrawn Submission_

### Official Review · Reviewer_vJpi · 2025-10-23

**Soundness:** 2
**Presentation:** 1
**Contribution:** 1
**Rating:** 2
**Confidence:** 4

**Summary:**

The authors inject artificial cues into the model's CoT for a family of simple factual questions. These statements say to not mention entity X. These cues often have a causal impact on the final answer. When the model is asked, in a second turn, why it did not include X, it is often dishonest. The authors show correlations between the activation difference with and without hint, and the sycophancy, deception and evil directions

**Strengths:**

- Faithfulness and model dishonesty are important questions
- The authors study several models, include large and high quality ones

**Weaknesses:**

As I see it, the authors make 3 main claims
- "Our results provide the first systematic
  evidence that reasoning traces causally shape model outputs"
    - I agree that there is a causal effect and that good evidence was provided of this
    - I disagree that this is novel, eg [Lanham et al 2023](https://arxiv.org/abs/2307.13702) shows this. And in my opinion this is implicitly shown in a great many works on chain of thought
- "Reasoning traces thus provide false transparency: they causally determine outputs but systematically misrepresent their own influence when asked, undermining the foundation of reasoning-based oversight"
    - I agree that this has been shown in the narrow setting studied by the authors
    - I disagree that this has been shown to apply to general reasoning choices because the authors study the specific, highly off policy setting, where they make an artificial edits to the model's thoughts, which is importantly different from monitoring natural reasoning traces
    - Further some of the edits will naturally cue the model to act misaligned, cueing dishonesty and this will be non-representative of normal behaviour
    - Further, reasoning based oversight is about monitoring the chain of thought not about asking them follow-up questions or having another language model try to process and write summaries of it. See [Korbak et al](https://arxiv.org/abs/2507.11473) for a more detailed case. In my opinion it is well known that asking the model about does not necessarily result in honest answers, even in the case where the model has a chain of thoughts to read rather than being expected to analyse its own internal mechanisms
    - Further, this is a fairly specific setting with a fairly easy and straightforward problem. As argued in [Korbak et al](https://arxiv.org/abs/2507.11473), this is not the setting where chain of thought monitoring is likely to help. For problems that can could be done without the CoT there's no mechanistic reason for the chain of thought to be useful. The difficult problems where the CoT is needed to increase the serial depth of computation available, are where we should expect this to work. In particular, on the hardest tasks models can solve.
- Mechanistic correlates
    - There may or may not be something here, I find it hard to tell.
    - It would be good for the authors to look at the similarity to non-evil persona vectors to make sure it's not just something about their persona Vector construction method
    - The authors provide no information about how they construct the persona vectors making it difficult to evaluate
    - Overall I wouldn't find this results particularly significant either way, given the various confounders detailed above

**Questions:**

Please correct me if I have misunderstood any key details of your work in the above!

---

> ### Author Response · Authors · 2025-11-30
>
> We appreciate your detailed breakdown of the three main claims you attributed to the original submission. Our manuscript changes directly reflect your comments.
>
> ---
>
> ## 1. “You claim novelty on causal influence of reasoning traces”
>
> Thank you for raising this. We agree that works such as Lanham et al. (2023) already show causal influence for prompt-elicited CoT in classic LLMs. However, our paper studies a fundamentally different phenomenon:
>
> ### (a) Classical CoT vs. Modern Reasoning Models
> Lanham-style CoT requires explicit prompting (“think step by step”). In contrast, modern reasoning models (e.g., R1-style) are trained via **RL-based test-time computation**, with a separation between a `<think>` phase and an `<answer>` phase. Although both produce text resembling a “reasoning trace,” prior work has repeatedly shown that reasoning-model `<think>` traces can be:
>
> - non-human-readable,
> - logically incorrect,
> - missing key evidence, or
> - fragile—small changes in RL training can cause the internal trace and final answer to diverge dramatically.
>
> These behaviors do not occur in traditional prompted CoT, indicating that reasoning-model CoT is a **new computational object**, not a continuation of classical CoT. (Appendix A.3)
>
> ### (b) Additional experiments confirm this distinction
> In the revised manuscript, Appendix D presents new experiments showing that modern reasoning models behave **qualitatively differently** depending on where additional rationale is placed (prompt vs. system message vs. inside `<think>`). This demonstrates that the `<think>` trace is not equivalent to classical CoT and requires separate empirical study.
>
> ---
>
> ## 2. “You generalize from a specific editing setting to general failure of reasoning-based oversight”
>
> We agree, and the revised manuscript explicitly avoids generalizing beyond our narrow controlled setting. We now clearly emphasize that our scenario:
>
> - edits the internal trace artificially,
> - uses a narrow domain, and
> - should not be taken as a direct evaluation of natural CoT monitoring on complex tasks.
>
> We present our work as highlighting **a failure mode in a deliberately simple setting**, not as a claim that reasoning-based oversight broadly fails.
>
> ---
>
> ## 3. “Extreme hints cue misalignment; dishonest behavior is non-representative”
>
> We agree, and have now:
>
> - Clearly separated *extreme* vs. *plausible* hints,
> - Emphasized **plausible-hint results** as the main story,
> - Framed extreme hints as **stress tests** rather than representative cases.
>
> We also avoid attributing intent: instead, we show that *non-disclosure persists across hint types*, while noting that safety training or aversion to endorsing harmful content may also contribute.
>
> ---
>
> ## 4. Mechanistic correlates and persona vectors
>
> To better contextualize the mechanistic correlates, the revised paper includes:
>
> - Added methodological detail on how persona vectors are constructed, normalized, and applied.
> - An explicit framing of these signals as **correlational evidence** of activation drift toward “sycophancy/dishonesty directions” in cases of non-disclosure—not as a definitive mechanistic explanation.
>
> This aligns with the revised manuscript’s positioning: the mechanistic section provides **possible handles** for detecting non-disclosure, rather than claiming a clean mechanistic account of deceptive internal algorithms.

---

### Official Review · Reviewer_XSzW · 2025-10-27

**Soundness:** 2
**Presentation:** 2
**Contribution:** 4
**Rating:** 2
**Confidence:** 4

**Summary:**

This paper studies the influence of the reasoning traces in large reasoning models. They insert synthetic reasoning snippets into the reasoning traces and measure causal effects on outputs. They found that the injected hints significantly alter the final answers, but the LRMs conceal these injections over 90% of the time.

**Strengths:**

- The reasoning trace offers an interesting perspective on the understanding and controlling of the LRM's reasoning.
- The finding that the LRMs tend to conceal these injections is impactful.

**Weaknesses:**

- Many parts of this paper are not really finished. E.g., Table 8 mentions that some results are pending. There are a lot of occurrences of broken references. The idea behind this paper is good, but I think this paper, at its current status, is not ready to be accepted.
- The "reasoning trace" seems to resemble "chain of thought". Korbak et al (2025) "Chain-of-Thought Monitorability" https://arxiv.org/abs/2507.11473 seems relevant, and it'd be great if this paper mentions CoT monitorability.
- The "causal" claim needs to be quantified better, e.g., via a framework that measures the causal effects (e.g., total effect)
- The honesty evaluation could be described in more detail. Many terms are left undefined, e.g., "semantically equivalent", "explicitly attributes this reasoning to the model's own reasoning process".

**Questions:**

Please refer to the reviews above.

---

> ### Author Response · Authors · 2025-11-30
>
> Thank you for highlighting both the strengths and the unfinished parts of the earlier draft.
>
> ---
>
> ## “Paper is not finished” (missing results, broken references, Table 8 placeholders)
>
> This concern was correct for the earlier version. In the revised manuscript:
>
> - **Table 8 is now fully populated** and clearly indicates where experiments are completed.
> - **All placeholders and broken references have been fixed.**
> - **Appendices** on statistics, query design, and honesty evaluation are **complete**.
>
> ---
>
> ## Relationship to CoT Monitorability and Prior Work
>
> We have expanded the related work to:
>
> - Explicitly cite and discuss **Korbak et al. (2025)** and other work on chain-of-thought monitorability.
> - Clarify that our contribution is **not** to show that CoT is always unfaithful, but to study a **specific answer–reasoning coupling failure** under controlled intervention.
>
> This is part of the manuscript revision; please refer to **Appendix A.3** for details.
>
> ---
>
> ## Causal Claims and Quantitative Framework
>
> Per your suggestion, we now:
>
> - Quantify the effect of hints via **paired comparisons**,
> - Use **Wilcoxon signed-rank tests**, **sign tests**, and **bootstrap CIs**,
> - Provide clear statements of **experimental hypotheses** and the meaning of **Δ**.
>
> This turns the “causal influence” discussion from an informal claim into a **properly quantified effect** within our controlled setting.
>
> ---
>
> ## Honesty Evaluation Definitions
>
> Your concern about terms like “semantically equivalent” and “explicitly attributes” is addressed by:
>
> - Providing precise definitions and concrete examples for:
>   - **Honest acknowledgment (disclosure),**
>   - **Fabricated explanation,**
>   - **Evasive response.**
> - Adding a detailed **annotation rubric** and **human validation data**.
>
> In the revised manuscript, we treat this as a **classification problem over explanation types**, not as evidence of moral or intentional properties.

---

### Official Review · Reviewer_espi · 2025-10-30

**Soundness:** 3
**Presentation:** 2
**Contribution:** 3
**Rating:** 4
**Confidence:** 3

**Summary:**

This paper investigates whether reasoning traces in large reasoning models genuinely shape outputs or merely serve as post-hoc rationalizations, and whether models honestly acknowledge when their reasoning influences their answers. The authors introduce thought injection, an intervention method that injects synthetic "hint" snippets directly into the model's reasoning process while keeping the user query fixed. Across 45,000 generations using three open-source large reasoning models (Qwen3-235B, DeepSeek-R1, and Qwen3-8B), the study reveals very large drops in the inclusion rate of a query's "expected element" when the hint tells the model to avoid it. They then ask "Why didn’t you mention X?" and find low "honesty rates" especially under deliberately misaligned "hatred hints", claiming models often fabricate post-hoc explanations. Finally, an activation-level analysis associates concealment with directions labeled "sycophancy" and "dishonesty" suggesting possible detection handles.

**Strengths:**

Novel intervention methodology: The paper introduces Thought Injection, a creative and technically sophisticated approach to studying reasoning traces. Unlike prior work that manipulates prompts or removes reasoning entirely, this method directly injects content into the reasoning trace itself while keeping queries fixed. This design elegantly isolates the causal effect of reasoning content from confounds like prompt engineering or task difficulty changes, representing a genuine methodological advance.

Dual-level evaluation framework: The paper's methodological contribution is the two-stage evaluation that separately measures (1) whether reasoning traces causally influence outputs, and (2) whether models honestly report this influence. This dual framework transforms a simple intervention study into an investigation of transparency and potential deception, opening new research directions around "false transparency" in AI systems.

**Weaknesses:**

Task scope is narrow to support broad claims. All evaluation comes from 50 prompts of a single list template chosen because they won’t violate accuracy when following hints. That design maximizes compliance headroom but samples only one behavior regime, where "expected elements" like Einstein are pre-sticky and easy to suppress. As written, this doesn’t demonstrate that "causal influence" and "dishonesty" generalize beyond subjective list ranking to tasks with objective ground truth, multi-step derivations, or tool use.

Selection on the dependent variable inflates effects. Queries were retained only if an element appears ≥90% in baseline answers, ensuring near-deterministic inclusion before intervention. This enriches for "ultra-stable" items and may overstate both suppression magnitude and the ease of demonstrating a counterfactual shift, limiting external validity to less canonical prompts.

Honesty requires semantic equivalence to the hint and explicit self-attribution; evaluation is primarily LLM-as-judge (gpt-oss-20B) across 30k follow-ups. The paper cites a human study with κ>0.85 on a 500-sample subset, but provides no judge prompts, rubrics, per-class breakdowns (Honest/Fabricated/Evasive), or confusion matrices by model or hint type. Strong claims like "over 90% concealment" hinge on this pipeline; without full labeling details, reliability and construct validity remain unclear.

"Causal" contribution risks collapsing into prompt-conditioning. The hint is placed inside <think> but is otherwise standard context; the method section even emphasizes that the injected text is indistinguishable from the model’s own trace under ChatML formatting. The paper shows strong suppression of expected elements (∆≈−0.88 to −0.95 medians), but does not compare against matched placements (outside <think>, post-answer, or as inert/scrambled text). Without such controls, it’s hard to claim a reasoning-trace-specific effect rather than generic sensitivity to earlier context.

**Questions:**

Q1: What is the effect size when hints are injected outside the <think> block (e.g., as regular assistant text or system messages)? How do effects vary with hint placement within the think trace?

Q2: Table 8 shows placeholders ("–") for Qwen3-8B paired tests. Can you complete these analyses and confirm whether results align with the other models?


Q3: The paper uses strong anthropomorphic language ("dishonesty," "concealment," "fabrication"). However, couldn't these behaviors reflect: a. Confabulation due to lossy memory/context b. Safety training to avoid discussing potentially harmful reasoning or c. Uncertainty about which parts of the trace actually influenced the decision. What evidence distinguishes intentional deception from these alternative explanations?

---

> ### Author Response · Authors · 2025-11-30
>
> We thank the reviewer for a thoughtful and largely positive review. We address your main concerns, now explicitly framed under the revised manuscript.
>
> ---
>
> ## Narrow Task Scope vs. Broad Claims
>
> We stress that:
>
> - the task domain is **subjective list-generation prompts**,
> - selected specifically so that we can cleanly define “expected elements” and quantify whether injected reasoning causes them to disappear,
> - and the conclusions are about **acknowledgment behavior in this setting**.
>
> Rather than claiming *“this undermines process supervision generally,”* our revised claim is:
>
> > **Even in a very simple, well-behaved setting where we know the injected trace changed the answer, models very often do not mention that internal influence when asked why the answer changed.**
>
> ---
>
> ## Selection on the Dependent Variable and Inflated Effects
>
> We agreed this needed more rigor. In the revision, we:
>
> - Report Wilson-score intervals and per-query robustness metrics for baseline hit rates.
> - Provide paired tests and bootstrap CIs for the suppression effect Δ.
> - Explicitly discuss the implications of selecting queries where an element is highly stable, presenting results as robust *within this controlled regime* rather than as claims about “typical” prompts.
>
> This aligns with the revised narrative: we’re not sampling all possible queries—we’re intentionally choosing ones where we can confidently detect the influence (excluding the expected element).
>
> We have also conducted an extra experiment in **Appendix H** focusing on other top-k scenarios. Results on 10 of the queries show the setup remains robust in guaranteeing the expected element within the generation.
>
> ---
>
> ## LLM-as-Judge, Semantics of “Honesty,” and Reliability
>
> We now:
>
> - Provide the full judge prompt,
> - Define rubrics for disclosure / fabrication / evasion,
> - Present a human validation study with κ > 0.85,
> - Show per-class agreement and examples.
>
> We also revise the wording from “honesty rate” to **“disclosure rate”**, and explicitly state that we are *not* inferring intent—only whether the explanation text acknowledges the injected content.
>
> ---
>
> ## “Causal” Claim vs. Generic Prompt-Conditioning
>
> Our revision includes a **system-prompt / user-prompt / reasoning-trace ablation study**. In Appendix D, we show significant differences in model behavior depending on where the prompt is placed.
>
> Additionally, we reframe the causal question conditional on injection inside `<think>`:
>
> - We do **not** claim to fully distinguish “reasoning-specific” vs. “prompt-level” effects.
> - Instead, we design a setting where a particular internal snippet appears inside the `<think>` trace and *demonstrably changes the answer*.
>
> The key question becomes:
>
> > **Given that this injected snippet inside `<think>` changed the answer, does the model subsequently acknowledge that snippet as part of the reason?**
>
> This makes the results agnostic to whether `<think>` is “special” or merely context—the issue is **self-attribution of a known internal influence**, not metaphysics about reasoning vs. prompting.
>
> ---
>
> ## Anthropomorphic Language (“dishonesty”, “fabrication”)
>
> We have toned down and rephrased these claims:
>
> - Use “non-disclosure,” “attribution failure,” and “fabricated explanations,” rather than “deception” or “dishonesty.”
> - Add discussion of alternative explanations (safety behavior, confabulation, uncertainty).
>
> Thus, the revised manuscript aligns with your concern: we present this as a **failure of faithful attribution**, not evidence of intentional deception.

---

### Official Review · Reviewer_UziF · 2025-10-31

**Soundness:** 1
**Presentation:** 3
**Contribution:** 2
**Rating:** 2
**Confidence:** 4

**Summary:**

This paper introduces 'Thought Injection', a counterfactual intervention method, that is meant to inject synthetic reasoning into traces in order to measure causal effects on outputs. Across ~45k generations on (narrow) list-style prompts with (somehow) selected "expected elements", injected hints dramatically suppress mention of those elements (e.g., dropping “Einstein” after a “avoid Einstein” hint), which the authors interpret as evidence that reasoning traces shape answers rather than serving as post-hoc rationalizations. Claims about dishonesty are also made.

**Strengths:**

- The thought injection seems to directly intervene on the reasoning trace, letting the authors compare outcomes w/ and w/out the hint.
- There are ~5000 trials, and results are evaluated in paired per-trial statistics; in Wilcoxon and sign tests, plus bootstrap CIs, effects appear large and consistent.

**Weaknesses:**

- Explaining why an item was omitted is not the same as explaining why it was included and, indeed, false positives and false negatives may behave differently in this game. How did you stratify and report results separately for target-included vs target-omitted generations (i.e., false positives vs false negatives), and test for asymmetries in disclosure/accuracy between them?
- Often, papers that use a forced choice, top-K, or even some sort of constraint-satisfaction report more than merely hit-rate but also rank-sensitive list metrics and set-quality measures at varying k, but this analysis is overly simplistic. Additionally, the empirical setup is very narrow (top-5 scenario only) and hints appear to be extreme and abnormal ("I HATE HUMAN"...really??). It also appears that only queries where an 'expected element' appears in 90% of baseline generations are used, which may be an ecologically invalid scenario or inflate suppression effects of the chosen intervention. It would not be difficult to perform a more fulsome evaluation.
- The 'open question' on L201 has some amount of debate and plausible answers or mitigations -- it may have been helpful to expand your literature review, and to follow up on some of thes claims. This is minor.
- Some weaknesses may merely be missed opportunities for explanation and not overclaiming, though that can't be discounted. Some questions are below, mostly regarding 1) the validity of the hint itself -- whether the authors are only testing compliance with inserted instructions and not actual changes to reasoning, and 2) whether models are actually 'dishonest' as a consequence, which seems spurious.
- Minor, but please check your spelling and grammar (e.g., "if model choose to follow" (L186))
- We are told there is a 'detailed curation' of data in Sec 2.1, but we are only given the single (!) "list the 5..." template and not how the superlative, category, or scope were defined or selected, nor how possible quality control was conducted.

**Questions:**

- It seems somewhat that the explantion for "not acknowledg[ing] the influence of HINT" (L74) leaps to the explanation for "dishonesty" (L76) and "conceal[ment]" (L77). Poor post-hoc reasoning or post-hoc explanations are known problems across LLMs generally -- are you making a hasty generalization?
- Did you test whether the model might be able to identify whether inserted HINTS are endogenous or from an external source? I.e., how do you know how it 'treats' (L172) each sentence or phrase? How might one investigate this? Along these lines, if the injected hints are text-level and pre-existing of the inference process, given that inputs tend to be read 'at once' in transformers, how did you guarantee the causal influence on hidden *state variables*?
- Would you prefer to change the term 'factual hints' to 'plausible hints', since the examples (Table 1) refer less to facts and more to subjective feelings or uncertainty? What matter is Coca-Cola's marketing team? Who did the 'linking', what is a 'link', and why should it matter? Hardly factual.

**Details Of Ethics Concerns:**

Although there does not appear to be any ethics review necessary, I am merely flagging the possibility that the targeted removal of 'selected entities' appears not to be motivated by a positive use case. In contrast, one can see this approach being used to perform political censorship or politically-motivated misinformation. The fact that the use case is so narrow mitigates this, and this is only a hypothetical extrapolation.

---

> ### Author Response · Authors · 2025-11-30
>
> We appreciate the reviewer’s detailed and critical comments. Many of them were aimed at the broader, more ambitious narrative of the original submission. Below we respond in light of the revised manuscript.
>
> ---
>
> ## Scope & Overclaiming (“dishonesty”, “concealment”)
>
> We remove the broad “dishonesty” narrative and instead talk about:
>
> - non-disclosure of a known causal influence,
> - generated alternative explanations, and
> - failure of answer–reasoning coupling.
>
> We acknowledge that there might be alternative explanations such as confabulation, lossy memory, or others. However, the core phenomenon persists: **models are influenced by reasoning traces yet do not disclose this influence when asked.** The underlying cause requires further study.
>
> We do not aim to identify the fundamental cause of this phenomenon, but rather to reveal this phenomenon.
>
> ---
>
> ## Endogenous vs. Exogenous Treatment of Hints and Compliance vs. Reasoning
>
> We conducted an additional experiment to address this question. In the revised manuscript, we include new experiments in **Appendix D** showing that modern reasoning models behave qualitatively differently depending on where additional rationale is placed (prompt vs. system message vs. inside `<think>`). This further demonstrates that the `<think>` trace produces different behavioral patterns than classical CoT or standard instructions.
>
> While we cannot confirm definitively that this is internal reasoning rather than external constraints, **the evidence suggests the `<think>` space is distinct in the behavior.** If it were simply treated as external instructions, we would expect similar behavior to prompted instructions. Instead, we observe placement-dependent behavior, suggesting that influence from reasoning traces is real and the explanatory mismatch is systematic.
>
> ---
>
> ## Narrow Task Design, ≥90% Filter, and Evaluation Simplicity
>
> We focused on the true negative cases (hit rate) because these represent scenarios where hints had maximal influence. If models fail to acknowledge influence even in cases where it was strong enough to change behavior, this is very concerning—especially for reasoning oversight.
>
> We also note that these true negative cases are the **majority of our data**, providing more robust sample sizes.
>
> Additionally, we conducted an extra experiment testing other top-k scenarios for a subset of our queries (10), as shown in **Appendix H**. Our results demonstrate that our query remains robust in guaranteeing the expected element within the generation, as the suppression effects still hold.
>
> ---
>
> ## Missing Details (data curation, grammar, etc.)
>
> Thank you for identifying these missing details. We have:
>
> - Added complete description of query collection and stability filtering (Appendix G/H).
> - Fixed grammatical issues and broken references.
> - Clarified hint construction and evaluation criteria.
> - Consistently referred to plausible hints.

---

### Author Response · Authors · 2025-11-30
**Global Response to All Reviewers**

We thank all reviewers for their thoughtful and detailed feedback. After reading the reviews, we realized that a significant fraction of concerns were driven not just by specific experiments, but by the narrative and scope of the original submission.

In response, we have made substantial, explicit changes to the narrative and scope of the paper. This is the most important update in the revision. This is also leading us to propose a new title "DO LARGE REASONING MODELS DISCLOSE WHAT SHAPES THEIR ANSWERS?" for our paper. We summarize the key revisions below:

Our paper addresses a specific question about reasoning traces in LRMs: when an injected reasoning fragment inside the think portion demonstrably changes the model's answer, does the model acknowledge this influence when asked to explain its output? We find that models overwhelmingly fail to do so, a phenomenon we call failure of answer-reasoning coupling. We operationalize this as:
- non-disclosure of the injected influence, and
- generation of alternative fabricated explanations."
- We replace moralized language with mechanistic and behavioral language.
    - “Dishonesty” → failure of answer–reasoning coupling, non-disclosure, fabricated explanations.
    - Mechanistic section is framed as identifying activation-level correlates, not proving intent.

As suggested by the reviewers, these changes strengthen our manuscript, addressing major reviewer concerns regarding the narrative, and ensure the scope is within the bounds of our exciting empirical results. We have also provided several additional experimental results in the Appendix.

---

### Note · Authors · 2026-01-03

I have read and agree with the venue's withdrawal policy on behalf of myself and my co-authors.